# Alert Germ Infections: Chest X-ray and CT Findings in Hospitalized Patients Affected by Multidrug-Resistant *Acinetobacter baumannii* Pneumonia

Raffaella Capasso [1],*, Antonio Pinto [1], Nicola Serra [2], Umberto Atripaldi [3], Adele Corcione [4], Giorgio Bocchini [3], Salvatore Guarino [3], Roberta Lieto [3], Gaetano Rea [3], Giacomo Sica [3] and Tullio Valente [3]

1 Department of Radiology, CTO Hospital, Azienda Ospedaliera dei Colli, 80131 Naples, Italy; antonio.pinto@ospedalideicolli.it
2 Department of Public Health, University Federico II of Naples, 80138 Napoli, Italy; nicola.serra@unina.it
3 Department of Radiology, Monaldi Hospital, Azienda Ospedaliera dei Colli, 80131 Naples, Italy; umbatripaldi@gmail.com (U.A.); giorgio.bocchini@aziendadeicolli.it (G.B.); salvatore.guarino@aziendadeicolli.it (S.G.); roberta.lieto@ospedalideicolli.it (R.L.); gaetano.rea@ospedalideicolli.it (G.R.); giacomo.sica@ospedalideicolli.it (G.S.); tullio.valente@ospedalideicolli.it (T.V.)
4 Department of Translational Medical Sciences, Section of Pediatrics, University Federico II of Naples, 80138 Napoli, Italy; adele.corcione@unina.it
* Correspondence: dott.ssacapasso@gmail.com; Tel.: +39-081-706-2629

**Abstract:** Acinetobacter baumannii (Ab) is an opportunistic Gram-negative pathogen intrinsically resistant to many antimicrobials. The aim of this retrospective study was to describe the imaging features on chest X-ray (CXR) and computed tomography (CT) scans in hospitalized patients with multidrug-resistant (MDR) Ab pneumonia. CXR and CT findings were graded on a three-point scale: 1 represents normal attenuation, 2 represents ground-glass attenuation, and 3 represents consolidation. For each lung zone, with a total of six lung zones in each patient, the extent of disease was graded using a five-point scale: 0, no involvement; 1, involving 25% of the zone; 2, 25–50%; 3, 50–75%; and 4, involving >75% of the zone. Points from all zones were added for a final total cumulative score ranging from 0 to 72. Among 94 patients who tested positive for MDR Ab and underwent CXR (males 52.9%, females 47.1%; mean age 64.2 years; range 1–90 years), 68 patients underwent both CXR and chest CT examinations. The percentage of patients with a positive CT score was significantly higher than that obtained on CXR (67.65% > 35.94%, *p*-value = 0.00258). CT score (21.88 ± 15.77) was significantly (*p*-value = 0.0014) higher than CXR score (15.06 ± 18.29). CXR and CT revealed prevalent bilateral abnormal findings mainly located in the inferior and middle zones of the lungs. They primarily consisted of peripheral ground-glass opacities and consolidations which predominated on CXR and CT, respectively.

**Keywords:** *Acinetobacter* pneumonia; multidrug resistance; chest X-ray; chest tomography; ground-glass; consolidation



## 1. Introduction

Multidrug-resistant (MDR) bacteria belonging to alert pathogens are an important cause of many severe and difficult -to-treat infections which greatly increase the morbidity and mortality among hospitalized patients worldwide [1]. In recent years, the phenomenon of MDR pathogens has increasingly become a cause for serious concern regarding both nosocomial and community-acquired pneumonia [2,3]. The World Health Organization (WHO) has recently identified antimicrobial resistance as one of the three most important problems facing human health. *Acinetobacter baumannii* (Ab) is one of the most common and serious MDR pathogens that have been encompassed within the acronym "ESKAPE"—standing for *Enterococcus faecium*, *Staphylococcus aureus*, *Klebsiella pneumoniae*, *Acinetobacter*

*baumannii*, *Pseudomonas aeruginosa* and *Enterobacter* spp.—and has been designated as a "red alert" human pathogen [2,4–6].

Ab is an opportunistic Gram-negative pathogen intrinsically resistant to many antimicrobials, which is difficult to treat and is associated with nosocomial outbreaks worldwide. Increasingly common in the intensive care units (ICUs), it is implicated in ventilator-associated pneumonia (VAP), infections of soft tissue, of urinary tract, catheter-associated infections, and primary bacteremia [7]. Among this wide spectrum of infections, the most common is hospital-acquired pneumonia (HAP) and it is associated with mechanical ventilation in more than 80% of cases [2,4,8–10]. Ab pneumonia appears to be the most dangerous pathologic condition associated with Ab showing the highest mortality rates [2]. Several trials highlighted the importance of MDR *Acinetobacter* pneumonia (AP), reporting mortality rates near to 90%, especially in patients who required mechanical ventilation [10]. Typically, these bacteria will cause VAP, especially in critically ill patients; recent studies have shown that approximately 1–28% of COVID-19 patients with the active disease on ventilators will subsequently develop a superimposed Ab infection while on the ventilator [11]. Furthermore, Carbapenem-resistant Ab infections in COVID-19-positive patients admitted to the ICU were more frequent compared to those that were COVID-19 negative [12]. In hospitalized patients, the respiratory tract is an important site of colonization and it is the most frequent site of infection [6,8,13]. *Acinetobacter* colonization has been reported from the nares, nasopharynx, and tracheostomy sites [13]. However, unlike community-acquired pneumonia, accepted clinical criteria for pneumonia are of limited diagnostic value in definitively establishing the presence of HAP, especially VAP, and it can be difficult to discriminate between true infection versus colonization [3,8].

Imaging plays a crucial role in the detection and management of patients with pneumonia [14]. To diagnose it, plain chest radiography (CXR) is an important initial and inexpensive examination allowing a rapid detection of pulmonary abnormalities. According to the literature, CXR findings associated with Ab pneumonia are generally bilateral with diffuse or multilobar consolidations and pleural effusions are not uncommonly an associated abnormality [10,15,16]. However, many studies reported that chest-computed tomography (CT) is a more sensitive technique thanks to its excellent spatial resolution. Unlike chest radiography, chest CT provides cross-sectional images; therefore, the pattern and distribution of pulmonary processes can be appreciated more readily with chest CT than with conventional radiography [14,17]. However, CT implies patient transportation to the radiology department, radiation exposure and costs [18]. In the literature, it has been reported that there are few characteristic features indicative of causative organisms and so radiographic findings of pneumonia generally do not provide a specific etiological diagnosis [14,17].

To the best of our knowledge, in the literature, there have been reported some case report and series, but there is no previous report which focuses on the imaging findings of AP [11,18–20]. The aim of this retrospective study was to describe the imaging features on CXRs and CT scans in hospitalized patients with MDR AP, which can help to suspect an infection. The pulmonary distribution and extent of consolidation and ground-glass opacity (GGO) were mainly described, in addition to other chest imaging abnormalities.

## 2. Methods and Materials

Between January 2019 and March 2020—immediately before the COVID-19 pandemic diffusion in Italy—patients who tested positive for MDR Ab were identified by a search in infective department databases and selected.

### 2.1. Patients

Formal consent of the local institutional review board was not required given the study's retrospective nature. All patients gave their consent before imaging examinations. The study group included 94 patients. In all patients, microbiological diagnosis was established by isolation of Ab from bronchoalveolar lavage fluid (n. 51) and endotracheal

aspirate (n. 43). MDR was defined as isolates that were insusceptible to three or more of the following antibiotics: anti-pseudomonal penicillin, anti-pseudomonal cephalosporins, carbapenems, monobactams, quinolones, aminoglycosides, tetracycline, and trimethoprim/sulfamethoxazole.

Antimicrobial susceptibility of Ab isolates was determined using the disk-diffusion test [21–24]. Pneumonia was diagnosed when at least two or more of the following criteria were satisfied: (1) fever (body temperature > 38.3 °C); (2) leukocytosis (25% increase and >10,000/mm$^3$) or leukopenia (25% decrease and <5000/mm$^3$); and (3) purulent tracheal secretion. One of the following criteria must also be satisfied: (1) new and persistent infiltrates appearing on the chest radiograph, (2) the same microorganisms are isolated from pleural fluid and tracheal secretions, (3) a radiographic cavitation, (4) histological proof of pneumonia, or (5) positive cultures from the bronchoalveolar lavage ($\geq 1 \times 10^4$ colony-forming units/mL) [21,22,25].

### 2.2. Chest Radiographs

CXRs were obtained using conventional radiography with postero-anterior projection, or portable computed radiography at bedside with anteroposterior projection. We chose one CXR for each patient, and it was obtained at approximately the first time of organism isolation.

### 2.3. CT Scans

All examinations were performed with the patient in the supine position and with breath-holding following inspiration (Toshiba Aquilion 64 system, Toshiba Medical Systems, Otawara, Japan). Contrast medium administration and acquisition protocols varied according to clinical indications. Images were captured at window settings that allowed the lung parenchyma (width 1200–1600 HU; level 2500 to 2700 HU) and the mediastinum (width 350–450 HU; level 20–40 HU) to be viewed.

### 2.4. Imaging Evaluation

Two chest radiologists (with 10 and 23 years of experience, respectively) retrospectively and independently interpreted the chest images with a final finding reached by consensus when there was a discrepancy. One CXR and one CT obtained at approximately the first time of organism isolation were evaluated for each patient.

To assess the pattern of pulmonary alterations, the CXR and chest CT images were analyzed with primary focus on parenchymal lesions and then their extent and distribution. Parenchymal lesions were distinguished in ground-glass opacity (GGO), consolidation, and nodular opacities (Figure 1). GGOs were defined as hazy areas of increased opacity or attenuation with preservation of bronchial and vascular markings. Consolidation was defined as homogeneous opacification of the parenchyma obscuring the underlying vascular structures. Nodular opacities were defined as focal round opacities (diameter < 3 cm). Other ancillary findings such as pleural effusion, lymphadenopathy (defined as a lymph node > 1 cm in short-axis diameter), interlobular septal thickening, bronchial wall thickening, and bronchiectasis were evaluated.

The extent and distribution of each abnormality were also analyzed. Whether the abnormal findings were located unilaterally or bilaterally was assessed. If the primary lesion was predominantly located in the inner third of the lungs, the disease was classified as having a central distribution. A peripheral abnormality was considered as an abnormality located in the outer third of the lungs. Furthermore, zonal predominance was classified according to three areas: upper (above the carina), middle (below the carina and above the inferior pulmonary vein), and lower (below the inferior pulmonary vein) (Figure 1).

The radiographic and CT findings were graded on a three-point scale: 1 as normal attenuation, 2 as ground-glass attenuation, and 3 as consolidation. The CT scans were scored on the axial images.

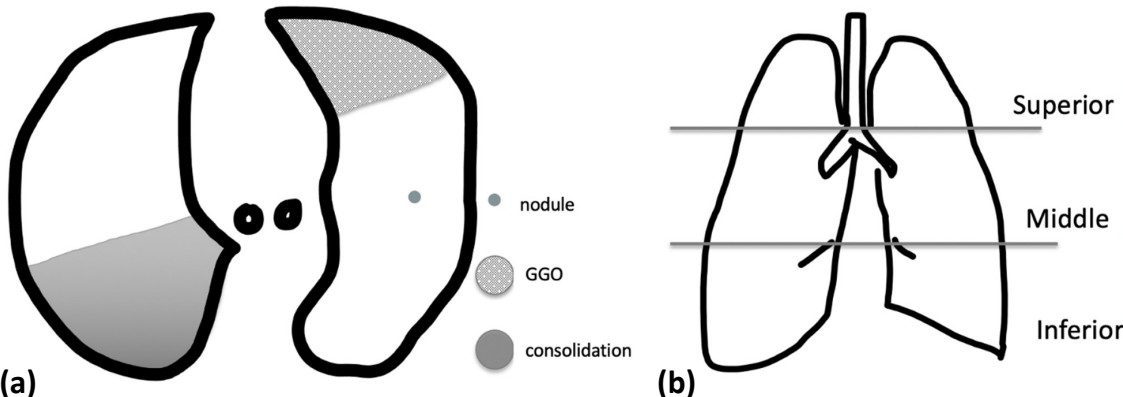

**Figure 1.** (**a**) Schematic representation of nodule, ground-glass opacity (GGO) and consolidation on an axial image; (**b**) schematic coronal representations of zones: superior ones above the carina, middle zones between the carina and the inferior pulmonary vein, and lower zones which are below the inferior pulmonary vein.

For each lung zone, with a total of six lung zones in each patient, the extent of disease was graded using a five-point scale: 0, no involvement; 1, involving, 25% of the zone; 2, 25–50%; 3, 50–75%; and 4, involving >75% of the zone. The five-point scale of the lung parenchyma distribution was then multiplied by the radiologic scale described above (Figure 2). Points from all zones were added for a final total cumulative score, with value ranging from 0 to 72.

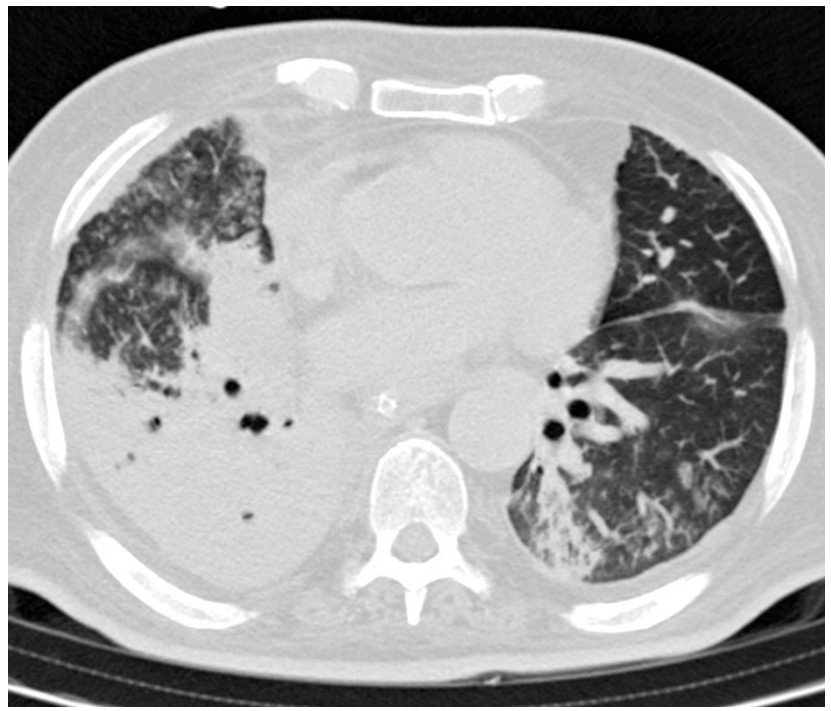

**Figure 2.** A sample scoring on an axial CT image of a 66-year-old man demonstrates a total score of 15, calculated as 3 (consolidation) × 3 (50–75% distribution in the right zone) + 2 (ground-glass opacity) × 2 (25–50% distribution in the left zone) + 2 (ground-glass opacity) × 1 (<25% distribution in the right zone).

## 2.5. Statistical Analysis

The statistical analysis was performed by Matlab statistical toolbox version 2008 (MathWorks, Natick, MA, USA) for Windows at 32 bit. The statistical tests performed were

the Student *t*-test, chi-square test with Yates correction and the McNemar's exact test and they were considered significant with *p*-value < 0.05.

## 3. Results

Among 94 patients who tested positive for MDR Ab and who underwent CXR (males 52.9%, females 47.1%; mean age 64.2 years; range 1–90 years), 68 patients underwent both CXR and chest CT examinations (Table 1).

**Table 1.** Demographic and imaging findings on CXR and CT examinations.

| | Number of Patients |
|---|---|
| Patients | 94 |
| Male | 53 |
| Female | 41 |
| Mean age (years) | 64.2 |
| **CXR** | 94 |
| GGO | 27 |
| Consolidation | 17 |
| GGO and consolidation | 20 |
| *Anatomic sides involved* | |
| Monolateral | 31 |
| Bilateral | 33 |
| *Pleural effusion* | 45 |
| Bilateral | 24 |
| *Involved zone* | |
| Superior | 27 |
| Middle | 44 |
| Inferior | 49 |
| *Score (mean $\pm$ SD)* | 15.06 $\pm$ 18.29 |
| **CT** | 68 |
| GGO | 5 |
| Consolidation | 26 |
| GGO and consolidation | 32 |
| *Anatomic sides involved* | |
| Monolateral | 9 |
| Bilateral | 54 |
| *Pleural effusion* | 43 |
| Bilateral | 34 |
| *Involved zone* | |
| Superior zone | 47 |
| Middle zone | 56 |
| Inferior zone | 61 |

**Table 1.** *Cont.*

| | Number of Patients |
|---|---|
| *Predominant distribution* | |
| Peripheral | 40 |
| Central | 4 |
| Central and peripheral | 19 |
| *Other findings* | |
| Nodular opacities | 10 |
| Cavitation | 4 |
| Bronchial wall thickening | 5 |
| Air bronchogram | 40 |
| Interlobar septal thickening | 17 |
| Bronchiesctasis | 10 |
| Lymphadenopathies | 18 |
| Pneumotorax | 2 |
| Pneumomediastinum | 1 |
| *Score (mean ± SD)* | 21.88 ± 15.77 |

### 3.1. Clinical Features

All patients had respiratory symptoms. The most frequently presented symptoms were fever (95.5%) and cough (91.1%), followed by sputum (86.7%), dyspnea (85.2%) and chest pain (72%). Most (92.6%) of the patients showed rapid progression in their respiratory symptoms.

In our study group, 59 patients were admitted to an intensive care unit (ICU), including those directly admitted to the ICU and those who transferred to the ICU during hospitalization, and received mechanical ventilator assistance.

### 3.2. CXR Findings and Disease Distribution

CXR was able to reveal any abnormalities in 81 patients (Table 1). In our study group, 64 patients showed increased lung density; the predominant CXR findings consisted of GGOs (n. 27, 42.2%) and mixed pattern of GGO and consolidation (n. 20, 31.2%), followed by areas of consolidation (n. 17, 26.6%). Primary abnormalities were bilateral in 33 (51.6%) patients. Pleural effusion was found in 45 patients (47.9%) and was bilateral in 24 (53.3%) cases. Inferior lung zones were the most affected (40.8%), followed by middle (36.7%) and upper (22.5%) ones. Enlarged mediastinal lymph nodes were not seen on CXR. The mean CXR score calculated for the 68 patients who also underwent a CT scan was 15.1.

The two radiologists were discordant in their independent CXR imaging evaluations in 29 cases.

### 3.3. CT Findings and Disease Distribution

Chest CT revealed any abnormalities in all patients (Table 1). Sixty-three patients presented CT lung densitometric alterations: the mixed pattern of GGO and consolidation (32, 50.8%) was the most frequent finding, followed by consolidation (26, 41.3%). These abnormal findings were found bilaterally in 54 (85.7%) patients and unilaterally in 9 (14.3%) patients. They had predominant peripheral distribution (63.5%), and mostly affected inferior (n. 61, 37.2%) and middle zones (n. 56, 34.1%) (Figure 3). Nodules (10 patients), bronchial wall thickening (5 patients), interlobular septal thickening (17 patients), bronchiectasis (10 patients), and air bronchogram (40 patients) were also observed. Cavitary lesions were found in only four cases [26]. Pleural effusion was found in 43 patients and was

bilateral in 34 (79.1%) cases. The mediastinal lymphadenopathies were observed in 18 patients. Lymph node enlargement was observed at the paratracheal, tracheobronchial and subcarinal regions. Pneumothorax and pneumomediastinum were seen in only three cases on CT. The mean CT score was 21.9.

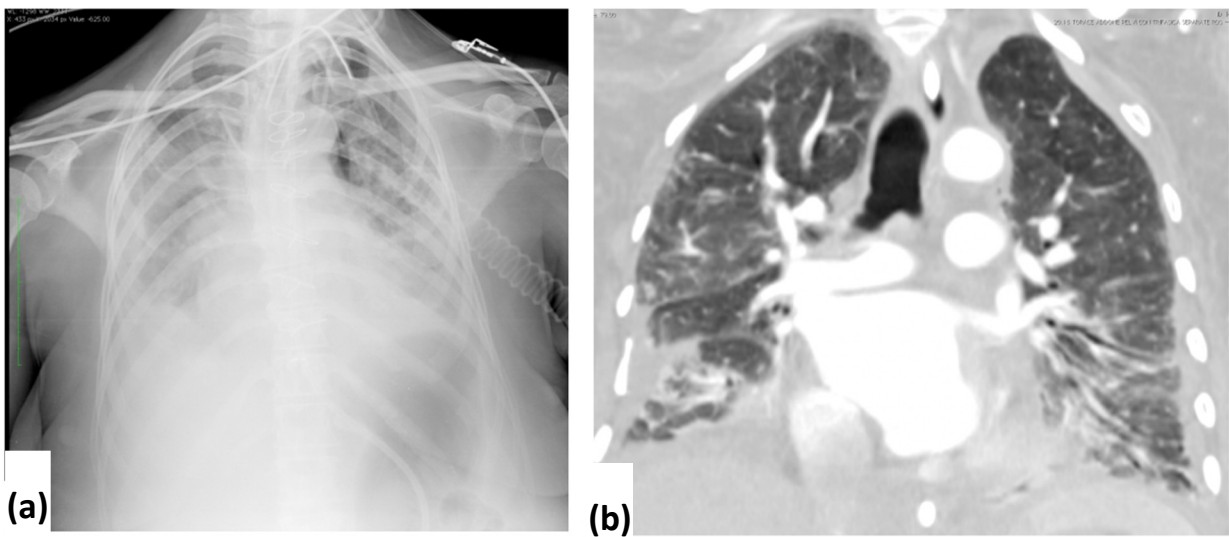

**Figure 3.** Portable CXR (**a**) and CT images (**b**, coronal) of a 55-year-old male patient: note the predominant involvement of the middle and lower lung zones with bilateral distribution.

No discrepancies were found in CT scan evaluations by the two radiologists.

*3.4. Statistical Tests*

Out of 68 patients who underwent both CXR and CT, in 4.41% (3/68) of cases, CXR and CT had the same positive score, in 2.94% (2/68) of the cases, CXR and CT had the same score equal to zero, in 63.24% (43/68) of cases, CT score was higher than CXR score, and in 29.41% (20/68) cases, CT score was lower than the CXR score. In detail, the proportion of patients with a positive CT score was 67.65% (46/68), while that with a positive CXR score was 35.94% (23/68). According to McNemar's exact test, the percentage of patients with a positive CT score was significantly higher than that obtained on CXR (67.65% > 35.94%, $p$-value = 0.00258). The results of the Student $t$-test showed that the CT score (21.88 ± 15.77) was significantly ($p$-value = 0.0014) higher than the CXR score (15.06 ± 18.29).

**4. Discussion**

Ab poses a significant threat to human health [12]. Several risk factors have been shown to be associated with *Acinetobacter* nosocomial infections. They include advanced age, immunosuppression, surgery, previous treatment with broad-spectrum antibiotics, use of invasive devices, burns, and prolonged hospital or ICU stays [2,10,20]. This is in agreement with the high percentage of our patients admitted to an ICU (90.4%), and with the advanced mean age observed in our study sample. The youngest patients included in the study were two children (1 and 7 years old, respectively), with physiological low immune defenses because of their immature-developing immune systems. Among patients younger than 30 years, one male (aged 19) was immunocompromised because he suffered from cystic fibrosis, one female (aged 25) was hospitalized for pneumothorax, one female (aged 25) suffered from pneumoperitoneum, one male (aged 26) was hospitalized for firearm injury, and one female (aged 19) presented septic shock due to vaginal tampons and was treated with extracorporeal membrane oxygenation.

As mentioned above, the most useful imaging modalities available for the evaluation of the patient with known or suspected pulmonary infection are CXR and CT [14]. CXR is routinely performed in hospitals for the initial management of patients with pneumonia

and the portable CXR still remains a mandatory modality in the diagnosis of ventilated patients with suspected pneumonia [17,27]. The possibility of its rapid and low-cost execution at the bed of the patient may explain why at least one CXR was available for all our patients, even in poor clinical conditions in which was not feasible to perform any CT examinations. However, poor-quality films, wrong radiographic technique and other clinical factors may further compromise the accuracy of CXRs which show problems with both sensitivity and specificity [8,13]. CXR does not always reflect the pathological findings of pneumonia due to the summation of opacity, and its reliability is limited by significant inter-observer variability in radiographic interpretation [7]. In our study, CT score evaluation generated no discrepancies between the two radiologists, while CXR imaging evaluation was characterized by interobserver variability (discrepancies in 29 patients).

Since ICU patients may also concurrently or otherwise have atelectasis, pulmonary infarction, pulmonary edema or acute respiratory distress syndrome, CXR is of limited value [15]. It has been reported that the overall radiographic specificity of a pulmonary opacity consistent with pneumonia is only 27% to 35% [17]. Indeed, the radiographic signs such as GGOs, consolidations and pleural effusion are nonspecific [8,15].

In our study, consolidation was the less frequent pattern found, and was little more frequently associated with ground-glass attenuation, while GGOs were the most frequent abnormalities observed on CXR. Bilateral involvement did not so much prevail on monolateral distribution of disease. These findings can be explained considering the low sensibility of CXR: although a normal CXR makes HAP unlikely, in one study of surgical patients, 26% of opacities were detected by CT scan but not by (portable) CXR [8].

Similar differences between CXR and CT sensibilities may validate the higher mean score obtained with CT examinations compared to the CXR score. Indeed, more minute findings can be recognized on chest CT, which furthermore allows a better evaluation of the extension of abnormalities [17]. The combination of consolidation and ground-glass attenuation was seen most frequently, nearly as frequent as consolidations alone, while GGO pattern was uncommon, unlike in the CXR evaluation results. This discrepancy may be due to higher sensibility of CT as discussed above, but also to the different number of patients who underwent CXR and both CXR and CT, and to the different times of execution of chest exams. It is likely that patients without consolidations on CXR did not perform CT, so GGOs discovered on CT were few, otherwise they underwent CT at a later stage, when GGOs converted to consolidations. Both CXR and CT, instead, revealed that the most involved zones were the inferior and middle ones, and this agreement follows from an evaluation of involved zones that was not stratified for lesion pattern, thus discrepancies between GGOs and consolidation observed on CXR and CT did not influence the assessment of lesion distribution. Moreover, because CT provides cross-sectional images, the pattern and distribution of pulmonary processes were much more readily appreciated than on CXR, revealing the predominant involvement of peripheral lung. Furthermore, CT allowed the assessment of ancillary findings, revealing underlying conditions or complications of infective disease.

The present study had several limitations: first, the different number of patients who underwent CXR and who underwent both CXR and CT; furthermore, the absence of follow-up evaluation and outcome analysis.

## 5. Conclusions

In this study, we evaluated CXR and chest CT scores assessed by determining the extent of patchy areas of GGO and/or consolidations in patients with MDR Ab pneumonia. CXR and chest CT scans revealed prevalent bilateral abnormal findings, mainly located in the inferior and middle zones of the lungs. They primarily consisted of peripheral GGOs and consolidations which predominate on CXR and CT, respectively. CT presented higher scores than CXR and appeared to be more suitable to depict consolidations, pleural effusions, and lymph node enlargement.

**Author Contributions:** Conceptualization, T.V. and R.C.; Methodology, R.C. and N.S.; Formal Analysis, N.S.; Investigation, T.V. and A.P.; Resources and data Curation, U.A. and A.C.; Writing—Original Draft Preparation, R.C.; Writing—Review and Editing, G.B., R.L., S.G. and G.S.; Visualization, T.V. and G.R.; Supervision, G.R. All authors have read and agreed to the published version of the manuscript.

**Funding:** This research received no external funding.

**Informed Consent Statement:** Informed consent was obtained from the patients.

**Data Availability Statement:** All data generated or analyzed in this work are original and there are no restrictions on data and material availability.

**Conflicts of Interest:** The authors declare that they have no conflict of interest. The manuscript has not been published before and is not under consideration for publication anywhere else. Publication is approved by all authors and, tacitly, by the responsible authorities where the work was carried out.

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
