# Peer review of "Alert Germ Infections: Chest X-ray and CT Findings in Hospitalized Patients Affected by Multidrug-Resistant Acinetobacter baumannii Pneumonia"

_tomography, doi:10.3390/tomography8030126_

Round 1

Reviewer 1 Report

The authors present the article entitled “Alert germ infections: chest X-Ray and CT findings in hospitalized patients affected by multi-drug-resistant Acinetobacter baumannii pneumonia.”. However, it is not possible to extend my recommendation for publication in the current form of the manuscript due to the following concerns:

The objective of the article is not clear. What is the novelty and contributions of the manuscript?

In perspective: Why was the mean reason that there is no reported about this topic?

The Introduction section is not impressive. I suggest studying in detail the state of the art to formulate adequately the novelty of the work.

Line 65: The correct term is COVID-19.

I recommend giving a brief introduction between section 2 and 2.1.

The manuscript presents some grouped citations. It is highly recommended that the authors give a brief mention of a particular contribution for each one.

The methodology section must be improved by showing how to reach the objective of the work.

The number of co-authors does not justify the impact of the work.

In general, my biggest concern is that the novelty is poor because the presented literature is too old. I recommend searching and study recent works to formulate the novelty of the work.

Table 1 is uncompressible.

The conclusion is poor. It reflects the novelty of the project.

Author Response

Response to Reviewer 1 comments.

First of all, we would thank the Reviewer for her/his comments which improve the scientific quality of the work.

You can find our response throughout the text in red color.

Open Review

(x) I would not like to sign my review report

( ) I would like to sign my review report

English language and style

( ) Extensive editing of English language and style required

( ) Moderate English changes required

(x) English language and style are fine/minor spell check required

( ) I don't feel qualified to judge about the English language and style

Yes      Can be improved        Must be improved       Not applicable

Does the introduction provide sufficient background and include all relevant references?

( )         (x)        ( )        ( )

Are all the cited references relevant to the research?

( )         (x)        ( )        ( )

Is the research design appropriate?

( )         (x)        ( )        ( )

Are the methods adequately described?

( )         (x)        ( )        ( )

Are the results clearly presented?

( )         (x)        ( )        ( )

Are the conclusions supported by the results?

( )         (x)        ( )        ( )

Comments and Suggestions for Authors

The authors present the article entitled “Alert germ infections: chest X-Ray and CT findings in hospitalized patients affected by multi-drug-resistant Acinetobacter baumannii pneumonia.”. However, it is not possible to extend my recommendation for publication in the current form of the manuscript due to the following concerns:

The objective of the article is not clear. What is the novelty and contributions of the manuscript?

The aim of the present study was to retrospective evaluate whether CXR and chest CT can help to recognize pattern of alteration attributable to Acinetobacter baumannii (Ab) pneumonia. It is reported that there are few characteristic imaging findings indicative of causative pathogen but there is no study that clearly illustrate CXR and CT findings of Ab pneumonia. This is the novelty, that previously nobody has analyzed chest imaging findings in patients with multi-drug-resistant Ab pneumonia. We think that may be interesting to know the radiologic pattern of chest alteration in affected patients. Although CXR and CT scan are not a sufficient diagnostic tool, familiarity with different imaging finding can be helpful for radiologists to support the diagnosis. In the last few years, lots of scientific paper described imaging finding of COVID-19 pneumonia clinching the same result (bilateral, multifocal, patchy GGO, with or without concurrent areas of consolidation, typically with a basal peripheral distribution); what was their novelty? SARS-CoV-2 itself? MDR infections represent a sanitary problem not less severe than COVID-19 pandemic.

In perspective: Why was the mean reason that there is no reported about this topic?

Most scientific works about multi-drug-resistant germs focus on clinical, laboratory and / or therapeutic rather than radiological aspects. It is easier to elaborate molecular data than radiologic finding which are not univocal and not very specific to generalize the conclusions. It is also difficult to select patients who satisfy inclusion criteria (multi-drug-resistant Ab pneumonia diagnosis, available CXR or chest CT -- Among imaging techniques, CXR is widely used despite its limited performance in VAP diagnosis. CT remains the gold standard. However, CT implies patient transportation to radiology department, radiation exposure and costs.) Our institution is the most important hospital for pulmonary and infective disease of the Southern Italy, so we could reach an adequate study sample of patient undergone to chest CT scan.

The Introduction section is not impressive. I suggest studying in detail the state of the art to formulate adequately the novelty of the work.

Introduction has also been modified according to Reviewer 2 comments.

Line 65: The correct term is COVID-19.

Corrected

I recommend giving a brief introduction between section 2 and 2.1.

Done

The manuscript presents some grouped citations. It is highly recommended that the authors give a brief mention of a particular contribution for each one.

Done

The methodology section must be improved by showing how to reach the objective of the work.

Done

The number of co-authors does not justify the impact of the work.

What to say

In general, my biggest concern is that the novelty is poor because the presented literature is too old. I recommend searching and study recent works to formulate the novelty of the work.

Recent works have been mentioned

Table 1 is uncompressible.

Modified

The conclusion is poor. It reflects the novelty of the project.

Had anyone ever described the pattern of changes on chest imaging in patients with multidrug-resistant Ab pneumonia before?

Reviewer 2 Report

This retrospective study studied the imaging features on Chest X-ray and CT scans in hospitalized patients with multi-drug resistant Ab pneumonia. The results showed that the percentage of patients with a positive CT score was significantly higher than that obtained on CXR. The proportion of patients with a positive CT score was 67.65%, while that with a positive CXR score was 35.94%.  
The presentation of this study is clear in Methods and Results but is poor in Discussion. There are bulks of general information that should be presented in the Introduction. There was almost no logic between paragraphs. Some paragraphs only have one sentence, which showed no relevance to the paragraphs above and below.    
Page 6, 1st paragraph in Discussion (lines 191-201): this is general information about the MDR bacteria and should be moved to Introduction.
 Page 7, lines 214-222: should move to Introduction.
Page 7, lines 223-224: This one sentence, one paragraph is strange here and should be removed or combined with the paragraph below.
Page 7, lines 225-230: should move to Introduction.
Page 7, lines 235-236: please rephrase “despite CXR im- 235 aging evaluations, CT scan evaluations created no discrepancies.”
Page 7, line 242: A one-sentence one-paragraph with no relevance in context.
Page 8, line 268-270: this sentence is Grammarly wrong and is difficult to understand: “By CT was possible, however, to better depict the localization of the lesions, thanks 268 to the axial visualization of the lungs without the summation of opacities which occur on 269 single projection CXR, and peripheral lung resulted the most involved.”
Page 8, line 273: limitations.
Page 8, lines 273-275: Grammarly wrong.

Minor:
Page 3, line 124: “The four-point scale” should be “the five-point scale”
Page 5, line 159” Change “68 patients undergone also to CT scan” to “68 patients also undergone CT scan”
Page 6, line 192: change “difficult to treat” to “difficult-to-treat”; add “,” before which.
Page 7, line 219, delete “for” after “unlike”.
Page 8, lines 280-281: “CT appeared to be ….”

Author Response

Response to Reviewer 2 comments.   First of all we would thank the Reviewer for her/his comments which improve the scientific quality of the work. You can find our response throughout the text in red colour.   Comments and Suggestions for Authors

This retrospective study studied the imaging features on Chest X-ray and CT scans in hospitalized patients with multi-drug resistant Ab pneumonia. The results showed that the percentage of patients with a positive CT score was significantly higher than that obtained on CXR. The proportion of patients with a positive CT score was 67.65%, while that with a positive CXR score was 35.94%.  
The presentation of this study is clear in Methods and Results but is poor in Discussion. There are bulks of general information that should be presented in the Introduction. There was almost no logic between paragraphs. Some paragraphs only have one sentence, which showed no relevance to the paragraphs above and below.    
Page 6, 1st paragraph in Discussion (lines 191-201): this is general information about the MDR bacteria and should be moved to Introduction.

moved

 Page 7, lines 214-222: should move to Introduction.

moved

Page 7, lines 223-224: This one sentence, one paragraph is strange here and should be removed or combined with the paragraph below.

combined with the paragraph below

Page 7, lines 225-230: should move to Introduction.

These sentences may be leaved there because they are useful to discuss some results related to the number of patient undergone to CXR and/or to chest CT

Page 7, lines 235-236: please rephrase “despite CXR im- 235 aging evaluations, CT scan evaluations created no discrepancies.”

rephrased

Page 7, line 242: A one-sentence one-paragraph with no relevance in context.

removed

Page 8, line 268-270: this sentence is Grammarly wrong and is difficult to understand: “By CT was possible, however, to better depict the localization of the lesions, thanks 268 to the axial visualization of the lungs without the summation of opacities which occur on 269 single projection CXR, and peripheral lung resulted the most involved.”

rephrased

Page 8, line 273: limitations.

modified

Page 8, lines 273-275: Grammarly wrong.

rephrased

Minor:
Page 3, line 124: “The four-point scale” should be “the five-point scale”

corrected

Page 5, line 159” Change “68 patients undergone also to CT scan” to “68 patients also undergone CT scan”

corrected

Page 6, line 192: change “difficult to treat” to “difficult-to-treat”; add “,” before which.

corrected

Page 7, line 219, delete “for” after “unlike”.

corrected

Page 8, lines 280-281: “CT appeared to be ….”

corrected

Round 2

Reviewer 1 Report

Although the authors responded poorly my comments, I can see that the manuscript has been improved

Reviewer 2 Report

Can be accepted.  But the presentation has a lot room to improve.  There are grammar errors.  But sentences are understandable, but was not expressed in clear way.